# Magnetic interference mapping of four types of unmanned aircraft systems intended for aeromagnetic surveying

Loughlin E. Tuck.[1], Claire Samson[1], Jeremy Laliberté[2], Michael Cunningham[1]

[1]Dept. of Earth Sciences, Carleton University, 1125 Colonel By Dr., Ottawa, ON, Canada K1S 5B6

[2]Dept. of Mechanical and Aerospace Engineering, Carleton University, Ottawa, ON, Canada K1S 5B6

*Correspondence to:* L. Tuck (loughlin.tuck@carleton.ca)

**Abstract**

Magnetic interference source identification is a critical preparation step for magnetometer-mounted unmanned aircraft

systems (UAS) used for high-sensitivity geomagnetic surveying. A magnetic field scanner was built for mapping the

low-frequency interference that is produced by a UAS. It was used to compare four types of electric-powered UAS

capable of carrying an alkali-vapour magnetometer: (1) a single-motor fixed-wing, (2) a single-rotor helicopter, (3) a

quad-rotor helicopter, and (4) a hexa-rotor helicopter. The scanner's error was estimated by calculating the root-mean-

square deviation of the background total magnetic intensity over the mapping duration; averaged values ranged

between 3.1-7.4 nT. Each mapping was performed above the UAS with the motor(s) engaged and with the UAS facing

in two orthogonal directions; peak interference intensities ranged between 21.4-574.2 nT. For each system, the

interference is a combination of both ferromagnetic and electrical current sources. Major sources of interference were

identified such as servo(s) and the cables carrying direct current between the motor battery and the electronic speed

controller. Magnetic intensity profiles were measured at various motor current draws for each UAS and a change in

intensity was observed for currents as low as 1 A.

## 1.    Introduction

Unmanned aircraft systems (UAS) are being used as an alternative method for traditional ground geomagnetic surveys

on the scale of <1km$^2$ (Eck and Imbach, 2012; Macharet et al., 2016; Parshin et al., 2018; Parvar et al., 2018; Versteeg

et al., 2010), <10km$^2$ (Kaneko et al., 2011; Koyama et al., 2013; Malehmir et al., 2017; Wood et al., 2016), and larger

(Anderson and Pita, 2005; Cherkasov and Kapshtan, 2018; Funaki et al., 2014; Pei et al., 2017; Wenjie, 2014). Their

ability to fly along tightly-spaced lines at low altitudes produces a higher resolution map than those produced by

typical manned aeromagnetic surveys. One of the main obstacles that impede the further acceptance of UAS in

aeromagnetic surveying is the interference generated by the UAS itself on the data recorded (Cherkasov and Kapshtan,

2018). Although magnetic interference is an issue that has been thoroughly investigated for manned aircraft (Coyle et

al., 2014; Hood and Teskey, 1989; Teskey, 1991), it is a more complex problem for UAS due to their smaller size and

shorter distances between the magnetic sensor(s) and source(s) of interference.

Magnetic interference from the UAS can be introduced from multiple types of magnetised sources and it is common

to investigate the impact of these sources before sensor installation and flight testing (Forrester, 2011; Jirigalatu et al.,

2020; Nelson, 2015; Parvar, 2016; Sterligov and Cherkasov, 2016). First, a combination of permanent and induced

magnetisation can occur in ferromagnetic materials where the strength of the latter magnetisation is a function of the

background field. These types of sources can be found in electric or fuel propulsion motor(s) and control servo(s)

(Cherkasov and Kapshtan, 2018; Forrester et al., 2014; Wells, 2008). Second, a field is produced from electric currents

flowing in electronic systems (Teskey, 1991). Examples of this type of source are electric motor(s), electronic speed

controller(s) (ESC), batteries as well as the leads that connect them (Tuck, 2019; Tuck et al., 2018). Third, induced

eddy currents in conductive materials can also produce interference (Fitzgerald and Perrin, 2015; Leliak, 1961). Apart

from multiple types of sources, their interference can also be divided into low-frequency and high-frequency

categories. An example of the former interference could be a magnetised fastener, whereas an example of the latter

could be a spinning magnetised propeller. During aeromagnetic survey, the low-frequency interference coincides with

measured frequencies of the magnetic geology (Hardwick, 1984) and therefore need to be dealt with differently.

 The issue of magnetic interference has been addressed using both software- and hardware-based approaches. In using

software, ferromagnetic interference (Naprstek and Lee, 2017; Noriega, 2011; Tolles and Lawson, 1950) and electric

current interference (Noriega and Marszalkowski, 2017) can be related to platform attitude and compensated for in

real-time or in post-processing. In using hardware, a straightforward approach is to increase the magnetometer-UAS

separation. One method has been to tow the magnetometer below the UAS at a distance where the interference

becomes negligible, often reported as >3 m (Cherkasov and Kapshtan, 2018; Koyama et al., 2013; Malehmir et al.,

2017; Parvar et al., 2018; Walter et al., 2018). This can introduce new issues such as location and heading error (Walter

et al., 2020), reduced flight stability, increased drag, and increased risk of impact damage to the magnetometer upon

landing (Kaneko et al., 2011). Additionally, these methods have not been demonstrated for fixed-wing UAS. Another option is to mount the magnetometer on a boom as an extension of the airframe's structure (Cunningham et al., 2018; Eck and Imbach, 2012; Funaki et al., 2014). This often results in a compromise between UAS interference and boom length; where a longer boom reduces interference but increases flight instability. Additional mitigation methods such as compensation using coils or rings (Leliak, 1961), shielding using permalloy (Leliak, 1961; Telford et al., 1990), demagnetisation using a degaussing coil (Camara and Guimarães, 2016; Tuck et al., 2019; Versteeg et al., 2007), optimal source positioning strategies (Forrester et al., 2014; Huq et al., 2015) or component replacement have been used. An effective way to deal with high-frequency interference is to sample at higher rates than the interference and remove with prefiltering (Pharr and Humphreys, 2010) and has been previously suggested for UAS magnetic interference mitigation (Versteeg et al., 2007).

For each method used to mitigate interference, it is desirable to identify the location and strength of magnetic sources. One way to achieve this and to assess the severity of the interference effects is through detailed magnetic interference mapping. Several magnetic interference investigations have been previously published (Cherkasov and Kapshtan, 2018; Forrester, 2011; Kaneko et al., 2011; Parvar et al., 2018; Sterligov and Cherkasov, 2016; Versteeg et al., 2007, 2010) but only a few include a detailed methodology for mapping the UAS. Forrester (2011) first mapped a 95 kg gas-powered fixed-wing UAS using a hand-held fluxgate magnetometer and identified three interference sources in order of severity: the servo(s) (50-100 nT at 0.55 m), the engine and engine assembly (60 nT at 0.55 m), and the avionics package (30 nT at 0.38 m). Forrester (2011) followed the mapping with individual testing of each component. Sterligov and Cherkasov (2016) mapped a 10 kg electric-powered flying-wing UAS using a planar surface as a measurement guide over top of the UAS and identified the major sources of magnetic noise as the electric motor (<800 nT), servos (<600 nT) and ferromagnetic elements (<300 nT). Parvar (2016) introduced three-dimensional mapping and isolated effects from the motor by calculating the difference in magnetic intensity when the UAS was powered on and off. He mapped a 5 kg electric powered hexa-rotor UAS and reported a 350 nT interference peak at 0.4 m. The experiment was repeated by Parvar et al. (2018) on a quad-rotor with a similar result. In both studies, the magnetometer was deployed 3 m below the UAS to mitigate interference. Finally, Tuck et al. (2018) mapped a 25 kg electric-powered fixed-wing UAS with the motor powered on and off using a non-magnetic test stand equipped with high-precision

satellite positioning. They measured intensities as high as 53.6 nT at a distance of 0.25 m behind the port side wing. Although the UAS in each study mentioned above vary in size and type, they each demonstrate high levels of interference that are not always symmetrically distributed across the UAS. In order for UAS to meet specified survey noise limits (e.g. <10 nT (Kaneko et al., 2011), <1 nT (Parvar, 2016), <2nT (Sterligov and Cherkasov, 2016), <2nT (Tuck et al., 2018)), interference sources often need to be identified and significantly mitigated.

Many different types of UAS are of interest for magnetic surveys and each with a unique magnetic signature that can evolve over time as modifications are made. This paper presents a robust and pragmatic method that will:

- map all types of UAS;

- identify sources of interference with sufficient positional accuracy to distinguish problematic areas;

- allow the UAS motor(s) to be engaged during mapping while keeping both the operator and the hardware safe;

- enable multi-directional mapping to discriminate between induced and permanent effects;

- identify interference that results from electrical currents;

- use the magnetometer and recording system to be installed on the UAS.

The method is demonstrated on four different types of electric UAS capable of carrying a survey-grade alkali-vapour magnetometer: a single-motor fixed-wing (FW), a single-rotor helicopter (SRH), a quad-rotor helicopter (QRH) and a hexa-rotor helicopter (HRH) UAS (Fig. 1, Table 1). The resultant interference maps are both a demonstration of the scanner on UAS, a comparison of the interference produced by different types of UAS employed for aeromagnetic survey, and a quick reference of typical problematic components. The mapping was performed with the motor(s) engaged at a single current. As a complement to each mapping, interference profiles were collected at different motor current draws to illustrate the impact of amperage on the magnetic signature of the UAS.

## 2. Magnetic Scanner

A scanner was designed and built to accurately map the magnetic interference of a UAS indoors while allowing the operator to remain at a safe distance while the UAS was in operation (Fig. 2). The scanner was also used in a previous study to map an unmanned ground vehicle for magnetic surveying (Hay et al., 2018). The scanner, constructed of low-susceptibility materials, moved a carriage transporting two magnetometer systems along an aluminium track above the UAS. The collection strategy over the UAS was chosen to be similar to that of an aeromagnetic survey to facilitate interpretation; a similar interpretation strategy used by Sterligov and Cherkasov (2016) and Jirigalatu et al. (2020).

The carriage was instrumented with the magnetic survey system intended for installation; a potassium-vapour total field (TF) magnetometer system (GSMP-35UAV, GEM Systems) powered by a 4 Ah lithium polymer (LiPo) battery and a triaxial fluxgate magnetometer (Mag649, Bartington Instruments) which recorded to a data acquisition system (DAS) (Raspberry Pi 3) powered with a 1.8 Ah LiPo battery. Both magnetometers were suspended on a rigid plastic boom 50 cm below the carriage and sample at a rate of 10 Hz. The TF magnetometer was used for mapping; the fluxgate magnetometer was used to measure the field direction.

The magnetic scanner was set up in a 6 m x 8 m laboratory and the length of the track was oriented along the magnetic north measured from the middle of the track. For each line, the carriage was towed along the track above the UAS using a timing belt and two 12 V stepper motors. The second stepper motor was added to avoid belt slippage and for additional torque to assure consistent speed of the carriage. The motors were operated by a control board located at one end of the track that delivered a maximum of 750 mA. The cart moved at a constant speed of 2.41±0.01 cm/s across the track translating to a measurement every 0.24 cm or 415 samples/m.

## 3. Method

The scanner was used to perform two tests for each UAS; (1) to produce an interference map at a constant motor current which is used to inform the spatial distribution of magnetic intensity and (2) to produce interference profiles

at various motor current draws which are used to inform how the magnetic intensity distribution changes with amperage.

### 3.1   Background removal

Measurement of the spatial and temporal variation of the background magnetic field within the laboratory is critical for indoor mapping. Interference at frequencies above the Nyquist frequency (5 Hz), such as 60 Hz electrical interference, are assumed to be aliased into the measurements. Previous work with the GSMP-35U magnetometer suggests that internal processing may apply filtering that could reduce interference aliasing (Tuck et al., 2018). Other magnetometers may have the ability to sample at higher rates that can accommodate anti-aliasing filters and reduce this interference (ex. Versteeg et al., 2007). Other, more complex interference sources that cannot be removed by simple methods, such as the variation in the inducing background vector, must be characterized before and during the mapping as it will have a major influence on the mapping error.

For each mapping, the measurements of the background magnetic intensity were made along the track length without the UAS present in order to:

a) Measure the spatial distribution of the background and provide a correction for isolating the anomalous field associated with the UAS. For example, the background for the FW mapping varied smoothly between an intensity of 52,100±2,500 nT (±5%), a declination of 0.6±50.6°, and an inclination of 85.2±3.1° (Fig. 3). The spatial distribution of the background was similar for each mapping.

b) Monitor variation of the background over time. The method assumes a minimal variation in background during the collection time. Line closure error ($CE_{line}$) was used to evaluate temporal variations between background lines by calculating the difference between $TF$ measurements at the north-end parking position for each set of sequential forward-return lines:

$$CE_{line} = \left| TF_{m,N} - TF_{m-1,1} \right| \tag{1}$$

where $TF_{x,y}$ corresponds to the $TF$ measurement number $n$ $\{n|n$ is an integer, $1 \leq n \leq N\}$ of the line number $m$ $\{m|m$ is an integer, $2 \leq m \leq M\}$. Lines with a $CE_{line}$ value greater than 5 nT were repeated. Similarly, map

closure error ($CE_{map}$) calculates the difference between measurements at the north end parking position for the whole mapping:

$$CE_{map} = \left|TF_{M,N} - TF_{1,1}\right| \tag{2}$$

The average (AVG) and standard deviation (STD) of $CE_{line}$ was 0.0 and 2.6 nT for the 8 mappings. $CE_{line}$ values were randomly distributed which was attributed to imprecise "parking" at line ends and small changes in the background. The AVG and STD of $CE_{map}$ was 2.2 and 5.6 nT for the 8 mappings.

c) Estimate the mapping error. Background lines were collected before, during, and after each mapping (Table 2). All codirectional background lines were compared to the first codirectional line of each mapping ($TF_1$). The mapping error is estimated using the root-mean-square deviation (RMSD) defined as:

$$mapping\ error = \sqrt{\frac{\sum_{i=1}^{N}(TF_{m',i} - TF_{1,i})^2}{N}} \tag{3}$$

where $m'$ is a codirectional line number. Over the 8 mappings carried out, the AVG and STD of the mapping error for all background lines was 4.2 and 1.1 nT.

Visual inspection of the residuals for each line, that is, the data remaining after the first codirectional line was subtracted, revealed coherent signals attributed to: (1) imprecise start and end line positions, or "parking" errors, (2) pendulum swing perpendicular to the track of the TF magnetometer due to air turbulence, (3) interference from the stepper motor apparent towards one end of the line, and (4) an irregularity in the middle of the track. The "parking" errors were apparent in the residuals as a low-frequency signal. This was produced by a positional shift of the line within the gradient of the laboratory. For most lines, this is the main contributor to the mapping error. Magnetometer pendulum effects, stepper motor noise, and the track irregularity were apparent in the residuals as higher frequency signal and were removed with a low-pass filter with a cut-off of 0.25 Hz in Sect. 4.2.

### 3.2 UAS scanning setup

Each UAS was fastened to a box made of non-ferromagnetic materials and positioned so that the top of the UAS was 30 cm below the magnetometer path. The QRH and HRH propeller blades were reversed to provide downward force when the motors were engaged. The magnetometer-UAS separation was chosen as a trade-off between safety and the mapping resolution which is a function of measurement distance from a source. Using the relationship between aliasing and the height to line-spacing ratio calculated for aeromagnetic surveys (Reid, 1980) and considering the limitation imposed on collection time by the UAS battery, a line spacing of 30 cm was chosen for the FW because of its larger dimensions, and of 10 cm for the other UAS (Table 3).

Each mapping was performed with the UAS in flight-ready configuration with the motor(s) engaged. The total UAS current drawn from the battery was measured using an ammeter. Mappings were performed with UAS as flight ready and with the motors engaged for two reasons. First, electronic systems on-board the UAS, such as the motors, avionics, transmitters and receivers, and other instrumentation, draw electric currents that will generate interference. The fields produced from these currents can influence the field produced by other ferromagnetic and conductive elements. Second, the high frequency interference produced by the magnets of an outrunner motor at high rotational speed is reduced significantly by what is assumed to be anti-aliasing filters in the GSMP-35UAV magnetometer (Tuck et al., 2018). The measurements of the filtered interference from the rotating motor are more representative of that experienced in flight and are independent of the orientation of the motor magnets when the motor is off. The motor controller of the multi-rotor UAS (QRH and HRH) was reconfigured so that each motor on the individual UAS had the same rotational speed and therefore a similar current draw.

Between each line, the UAS was moved perpendicular to the track length in equal increments described in Table 3 for full coverage. Data was recorded with the front of the UAS oriented to the magnetic north and then east to capture any dependence of the interference on the orientation of the background. One exception was the FW which could not be oriented eastwards because the space in the lab could not accommodate its large wingspan.

## 4. Results

### 4.1 Interference mapping

The interference maps for the FW, SRH, QRH, and HRH, under the conditions described in Table 3, are presented in two orthogonal orientations in Fig. 4, 5, 6, and 7, respectively. These figures show the magnetic interference associated with the UAS after the subtraction of the background.

The interference map of the FW, when powered with a 10 A current, exhibits two large dipolar anomalies in both the

north (top) and west (bottom) orientations which remain fixed to the airframe under rotation (Fig. 4). The anomalies are centered in the fore and aft of the UAS. The position of the fore dipole corresponds with the motor system in the nose of the UAS in the north ($north_{min}$=-127.4 nT) and in the west ($west_{max/min}$=+110.9/-106.3 nT). Its position overlies the motor battery, ESC, motor, and associated cables. The other dipole is located around the tail ($north_{max/min}$=+167.2/-121.1 nT and $west_{max/min}$=+161.2/-153.2 nT) and coincides with the location of 3 servos and the steel supports located

in the tail. The negative lobe of the tail dipole connects to a negative lobe associated with each wing, possibly associated with the flap and aileron servos (located at 120 cm and 75 cm from each wingtip, respectively) or the steel linkages that connect the servos to the moveable flight surfaces.

The SRH blades were removed for safety and the resulting maximum power draw by the motor was 1.5 A. The

interference of the SRH presents a negative single polar anomaly in both mapping orientations. The negative anomaly is not representative of induced interference. Due to its symmetrical signature, no conclusion could be drawn regarding intensity changes resulting from airframe rotation (Fig. 5). The interference minimum ($north_{min}$=-574.2, $west_{min}$=-566.8 nT) coincides with the centre mast, motor and servo batteries, motor, ESC, servos, and motor controller/receiver and associated cables. Since the large negative single pole was generated under low current conditions, it suggests that

the source was unrelated to the motor's electrical system. Instead, the anomaly could be from the four servos located around the centre mast or the magnetisation of ferromagnetic components also located in the centre mast.

The interference map of the QRH, when powered with a 10 A current, exhibits a positive single polar anomaly ($north_{max}$=+66.1 nT, $west_{max}$=+75.0 nT) which remains fixed to the airframe under rotation (Fig. 6). The interference

anomaly peaks at the centre of the body but displays some amplification along the conductive aluminum arms. The

anomaly does not follow one arm, forming a triangular shape. This lower interference in one arm could be a result of

different wire twisting or an issue with this particular motor. In general, the field is not associated with the motors but

appears to be from a single source located at the centre of the UAS where the battery, ferromagnetic fasteners in the

battery carrier, and the motor controller/receiver are located.


The interference map of the HRH exhibits a dipolar anomaly when powered with a 5 A current. In the mapping plane,

the dipole is predominantly negative (north$_{max/min}$=+103.4/-464.4 nT, west$_{max/min}$=+21.4/-470.2 nT) and remains fixed

to the airframe under rotation. The centre of the dipole corresponds with the centre of the UAS where the battery,

battery cables, and gimble servo are located. It does not coincide with the 6 motors or 6 ESCs that are located at the

end of each plastic arm.

### 4.2    Interference profiles

The interference profiles were recorded for each north-facing UAS at different motor current draws (Fig. 8). Each

UAS was positioned so that the magnetometer path ran directly through the centre of the UAS. The throttle was

adjusted between profiles using a remote transmitter and the UAS was not moved.

The SRH motor current draw could not exceed 1.5 A with the blades removed and therefore the interference

relationship with amperage could not be investigated. The FW, QRH, and HRH profiles change with increasing

current; changes are visible for currents as low as 1 A. For these three UAS, the greatest changes coincide with the

position of the battery and cables going to the ESC.

An interference profile without current-induced interference (0 A) can be calculated for each UAS (Fig. 8) by linear

extrapolation and represents the permanent and induced magnetisation (herein magnetisation interference). This

magnetisation interference was subtracted from each higher current profile leaving the current-induced interference

for each amperage (Fig. 9 (top) for the HRH). The minimum intensities (for FW) and the maximum intensities (for

QRH and HRH) of the current-induced interference are plotted with respect to current (Fig. 9, bottom). In each case,

the peak intensity had a linear relationship to current ($R^2$ = 0.998, 0.991, and 0.999, respectively) with slopes of -7.5, 2.8, and 10.4 nT/A, respectively. Since the interference remains fixed with the airframe under rotation (Sect. 4.1), the magnetisation interference appears to be largely permanent.


The separation of the interference profile provides new information that compliments the mapping shown in Sect. 4.1. For example, the apparent dipole observed in the mapping of the HRH was in fact two single poles from separate sources that are centred at different locations. The location of the magnetisation interference centre relates well to the location of the gimbal servo whereas the centre of the current-induced interference coincides with the cables from the

battery. Another example was the magnetisation interference in the QRH profile that, unlike the other three, cannot be attributed to a servo. Further investigation found a group of ferromagnetic fasteners located in the battery-carrying cage that may have become magnetised.

## 5. Discussion and conclusion

This paper presents a quick and pragmatic method for mapping the low-frequency magnetic interference sources of a UAS in a laboratory setting. In contrast to other interference investigations, this paper presents a method that allows the UAS to be powered and running while data is collected in a semi-automated fashion that increases the maps accuracy and safety. The mappings are in two directions in order to discriminate induced interference, and profiles measures interference at different current draws. This provides a more complete picture of the low-frequency magnetic

interference produced by UAS sources.

Locating and characterizing sources is a key first step to interference mitigation before magnetometer location selection or after platforms have been modified. The method proposed locates interference sources quantitatively by producing detailed maps. Their interpretation of character is akin to that of aeromagnetic survey maps used for locating

geological magnetic sources. When the sources are located and characterized, and the strength quantified, calculating the interference at any installation point can be done using field theory. An example of this is Sterligov and Cherkasov, 2016.

To produce interference maps, a scanner was built with the purpose of minimizing positional inaccuracies by utilizing stepper motors designed for printing applications. The mapping error was estimated by calculating the change in magnetic intensity of the background lines over the mapping time (AVG and STD RMSD of 48 lines over 8 mappings: 4.2 and 1.1 nT) and is small with respect to the large anomalies associated with the UAS. The largest contribution to the mapping error was the background subtraction and the result of lines with a high "parking" error within the magnetic gradient of the laboratory. This error was most prevalent on the edges of the FW mappings. The mapping error could be further reduced by mapping in an area of lower gradient or by programming exact line lengths into the stepper motor controller to reduce parking error. Shielding the stepper motors and reducing the pendulum swing would reduce the mapping error as well.

Four different types of UAS capable of carrying an alkali-vapour magnetometer were magnetically mapped using the scanner. For each mapping, the magnetic interference is measured at levels significantly beyond typical survey noise limits (Sect. 1) and therefore interference mitigation steps are warranted. For each system, the interference is a combination of both ferromagnetic and electrical current sources. Ferromagnetic sources are identified as differently oriented dipolar anomaly(ies) intersecting with the measurement plane. In most cases, the ferromagnetic elements are predominantly permanently magnetised where their dipolar orientations do not coincide with the downward pointing background vector; this type of field was only present in the QRH maps. These anomalies are centred on sources such as servo(s) which contain permanent magnets and ferromagnetic fasteners. As Ampère's law predicts, the interference produced by direct electronic current increases linearly with current through the cables between the motor battery and the ESC and is detectable for currents as low as 1 A. Currents of 40 A or more can be expected during flight for each of these UAS making current interference, without any mitigation, the dominant source of interference during flight.

Without using any mitigation strategies, the most effective way to remove current-induced interference is to locate the magnetometer outside the zone of influence of the interference sources. The QRH with no servos or moving flight surfaces produces the smallest magnetic interference signature with predictable permanent and current-induced interference contributions. A subtraction of the calculated magnetisation and current-induced interference from each

QRH interference profile leaves a residual peak <5 nT. This would represent a 93% reduction of the peak measurement of the 25 A interference profile. Based on these merits and the implementation of a short boom, it could potentially be a good choice for geomagnetic surveying. Alternatively, the larger FW exhibits low levels of interference on the wingtips before interference mitigation or compensation methods have been applied. The wingtips on the FW has the most potential for a low-interference installation.

**Author contribution**

L. Tuck planned, collected, processed and analyzed the experimental data and wrote the manuscript. C. Samson assisted in establishing the research objectives and provided extensive comments on the technical results and the manuscript. J. Laliberté assisted in establishing the research objectives and provided valuable comments on the manuscript. M. Cunningham assisted with the design and construction of the scanner, assisted with the diagram of the scanner (Fig. 2) and provided valuable comments on the manuscript.

**Competing interests**

The authors declare that there are no competing interests regarding the publication of this paper.

**Acknowledgements**

This work was supported by a Collaborative Research and Development grant from the Natural Sciences and Engineering Research Council of Canada (NSERC) to J. Laliberté, C. Samson and D. Feszty, sponsored by Sander Geophysics Ltd. Support was provided to L. Tuck via a NSERC Doctoral Postgraduate Scholarship and an Ontario Graduate Scholarship. The magnetometers were lent to the project as an in-kind contribution from GEM Systems Canada.

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

**Tables**

**Table 1: UAS specifications.**

| | Fixed-wing (FW) | Single-rotor helicopter (SRH) | Quad-rotor helicopter (QRH) | Hexa-rotor helicopter (HRH) |
|---|---|---|---|---|
| **UAS make and model** | 33% Scale Piper Pawnee (Hangar 9, 2021) | T-Rex 600E Pro (Align, 2021) | DYS D800 X4 (Hobby King, 2021) | S800 Evo (DJI, 2021) |
| **Electric propulsion system** | 1x Hacker Q80 -6L V2 7000 W 180 Kv, brushless outrunner motor (Hacker Motor USA, 2021) | 1x Eflite Heli 700, 700 W 500 Kv, brushless outrunner motor | 4x Elite 5008, 610 W 330 Kv, brushless outrunner motor | 6x 4114 Pro, 500 W, 400 Kv, brushless outrunner motor |
| **Batteries** | 12S-22Ah (motor), 4S-2.2Ah (autopilot), 3x 2S-5Ah (avionics) | 12S-10Ah (motor), 2S-1Ah (avionics) | 6S-16Ah | 6S-15Ah |
| **Electronic speed controller** | Castle Phoenix Edge HV 160 A | Castle Phoenix 120 A | 4x ESC 40 A | 6x ESC 40 A |
| **Maximum payload (kg)** | 5 | 4 | 6.5 | 4.3 |
| **Dimensions (cm)** | 330 wingspan/240 length | 21 x 116 | 80 x 80 | 80 x 80 |
| **Servos** | 7 servos - 4 wing, 3 tail (1 rudder, 2 elevator) | 4 servos (3 swashplate, 1 tail) | 0 | 1 gimble servo |


**Table 2: Statistics on background lines for each mapping.**

| Mapping | Line length (m) | Number of background lines | Mapping time (min) | AVG RMSD (nT) | STD $CE_{line}$ (nT) | $CE_{map}$ (nT) |
|---|---|---|---|---|---|---|
| FW-north | 2.9 | 6 | 138 | 4.9 | 4.0 | 13.9 |
| FW-west | 2.9 | 6 | 95 | 7.4 | 4.4 | 6.9 |
| SRH-north | 2.1 | 6 | 88 | 3.5 | 1.2 | 1.8 |
| SRH-east | 2.1 | 6 | 68 | 3.1 | 1.9 | 1.6 |
| QRH-north | 1.6 | 8 | 89 | 4.7 | 2.1 | 0.5 |
| QRH-east | 1.6 | 13 | 101 | 3.4 | 1.2 | 1.5 |
| HRH-north | 1.6 | 6 | 120 | 5.0 | 3.2 | 0.7 |
| HRH-east | 1.6 | 13 | 107 | 3.6 | 3.2 | 2.9 |

**Table 3: UAS mapping parameters.**

| UAS type | Motor speed (RPM) | Current (A) | Nominal voltage (V) | Line spacing (cm) |
|----------|-------------------|-------------|---------------------|-------------------|
| FW | 1300 | 10 | 45.6 | 30 |
| SRH | 1400 | 1.5 | 45.6 | 10 |
| QRH | 2100 | 10 | 22.8 | 10 |
| HRH | 2200 | 5 | 22.8 | 10 |


**Figures**

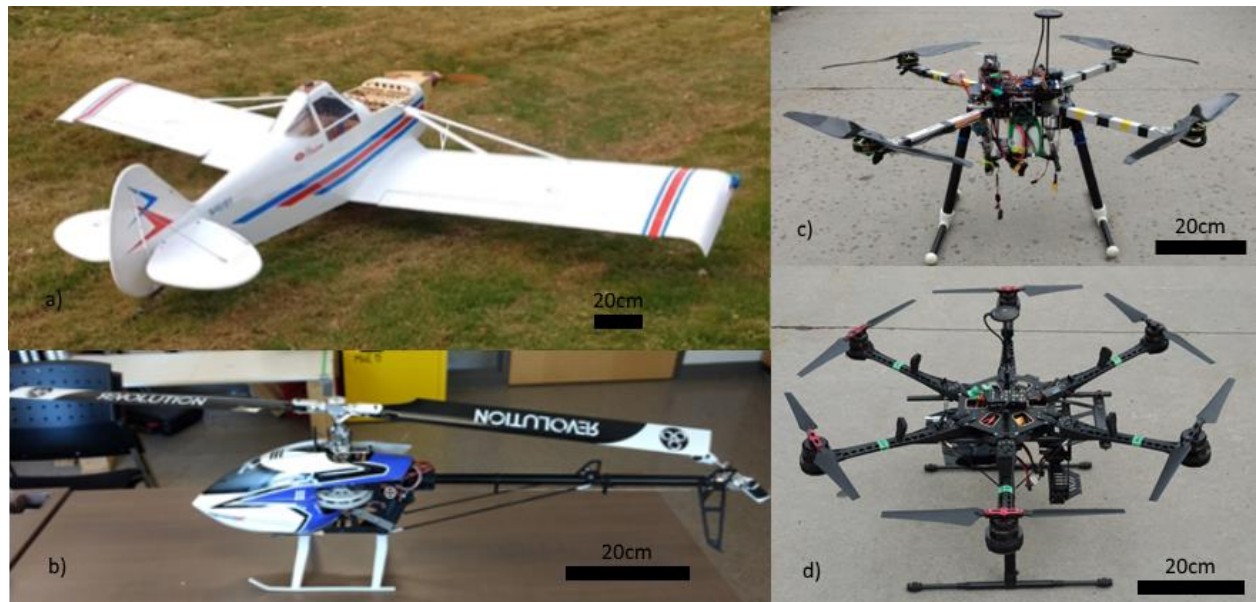

Figure 1: Photographs of the UAS investigated in this study. a) single-motor fixed-wing, b) single-rotor helicopter c) quad-rotor helicopter, and d) hexa-rotor helicopter.


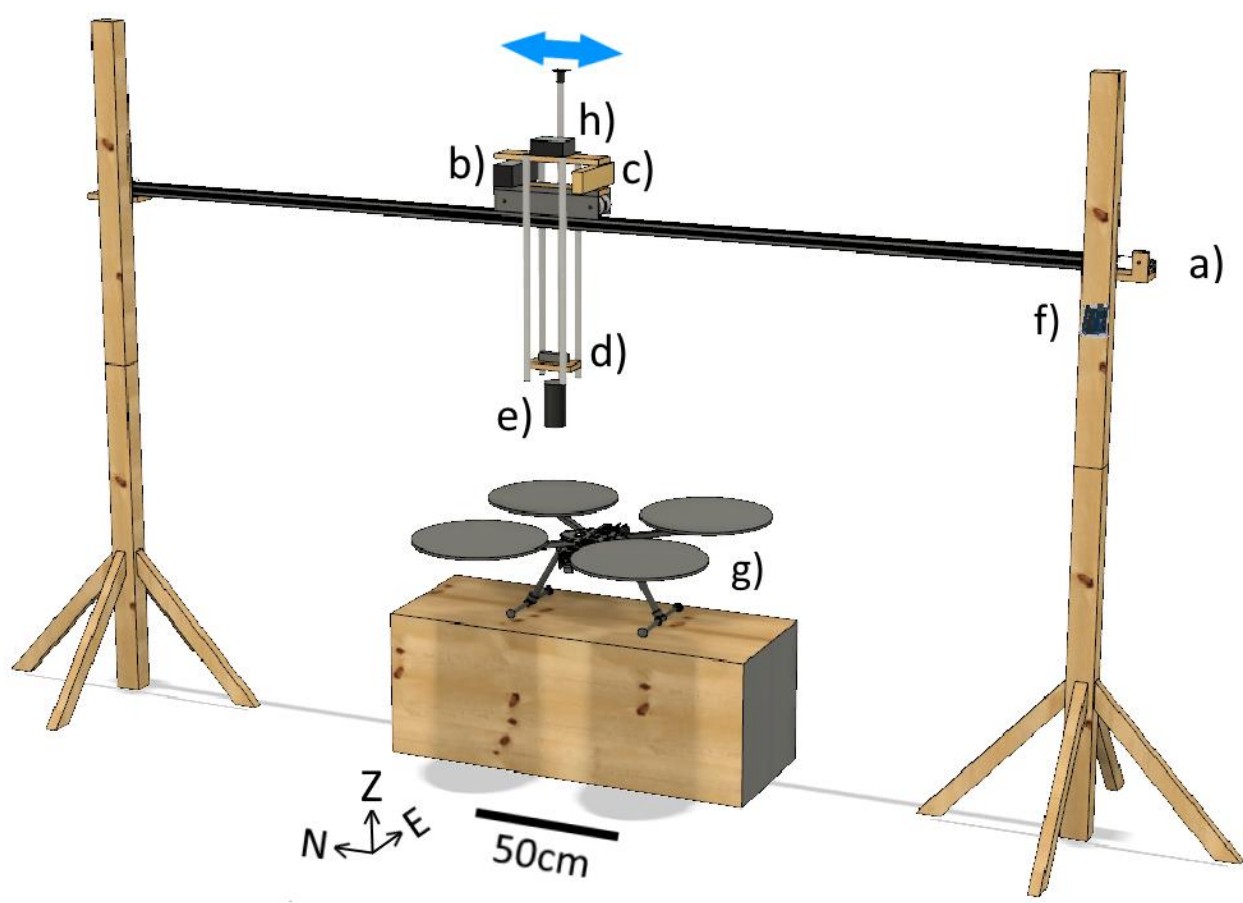

**Figure 2: Magnetic scanner composed of: a) two stepper motors, b) magnetometer battery, c) magnetometer DAS, d) triaxial fluxgate magnetometer, e) TF magnetometer, f) stepper motor control board, g) UAS, and h) fluxgate DAS. Cables are removed for simplicity.**


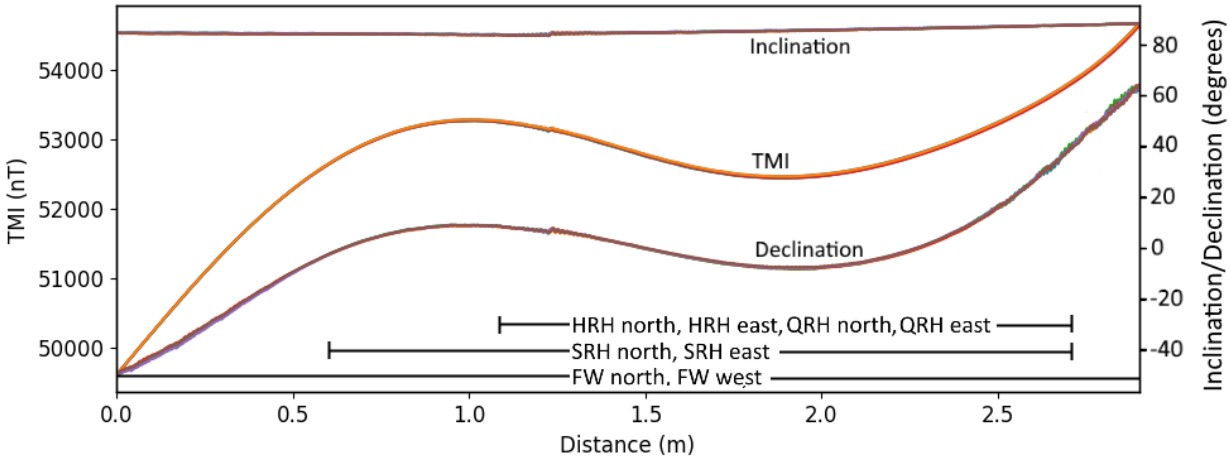

**Figure 3: The TMI, inclination and declination profile of 6 background lines collected during the FW north mapping. Repeat background lines appear superimposed at this scale. The location of the line lengths (table 2) used for each mapping is noted at the base of the plot.**


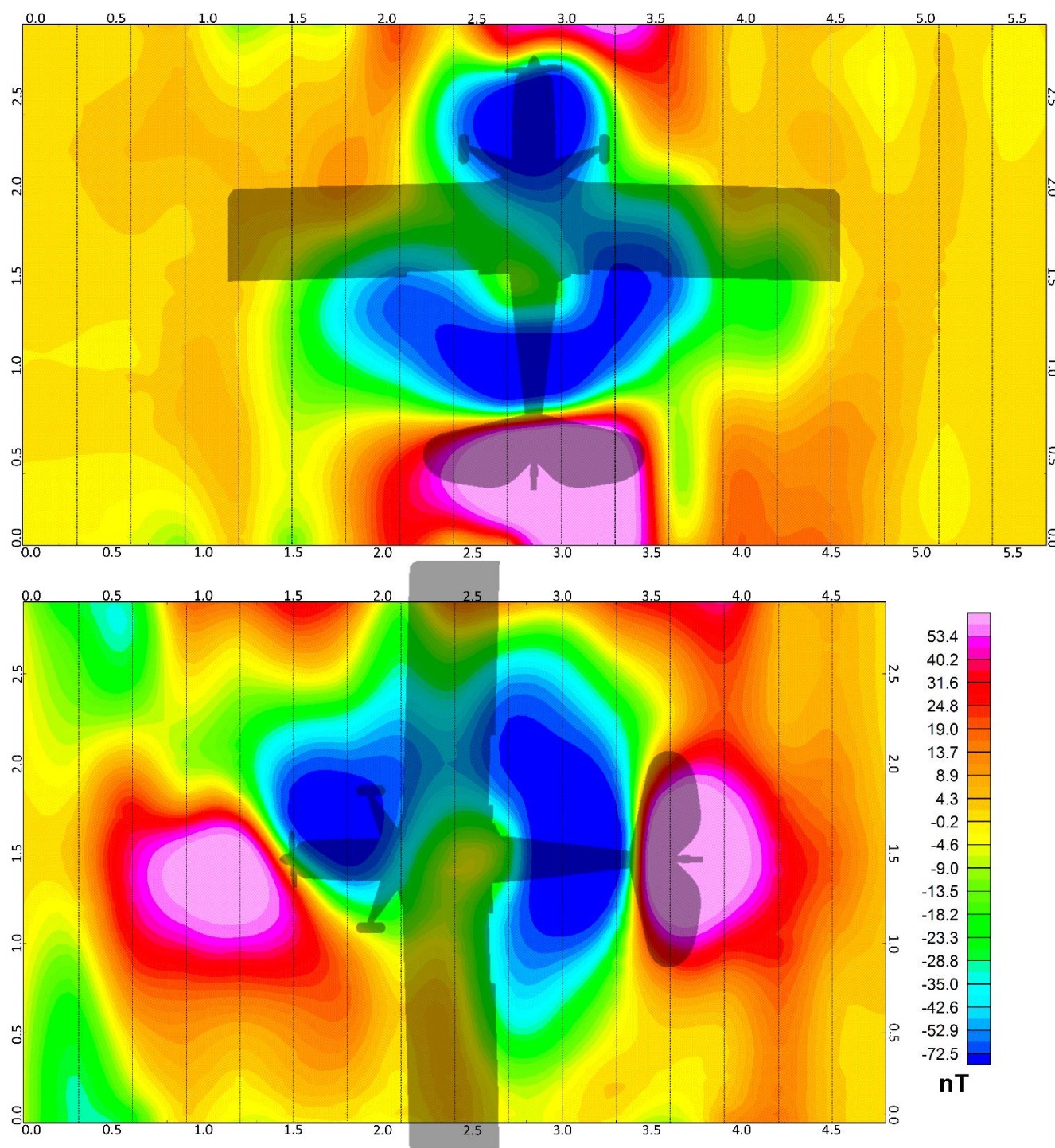

**Figure 4: Interference map of the FW facing magnetic north (top) and west (bottom). Border units are in metres. Edge effects are present on both maps.**


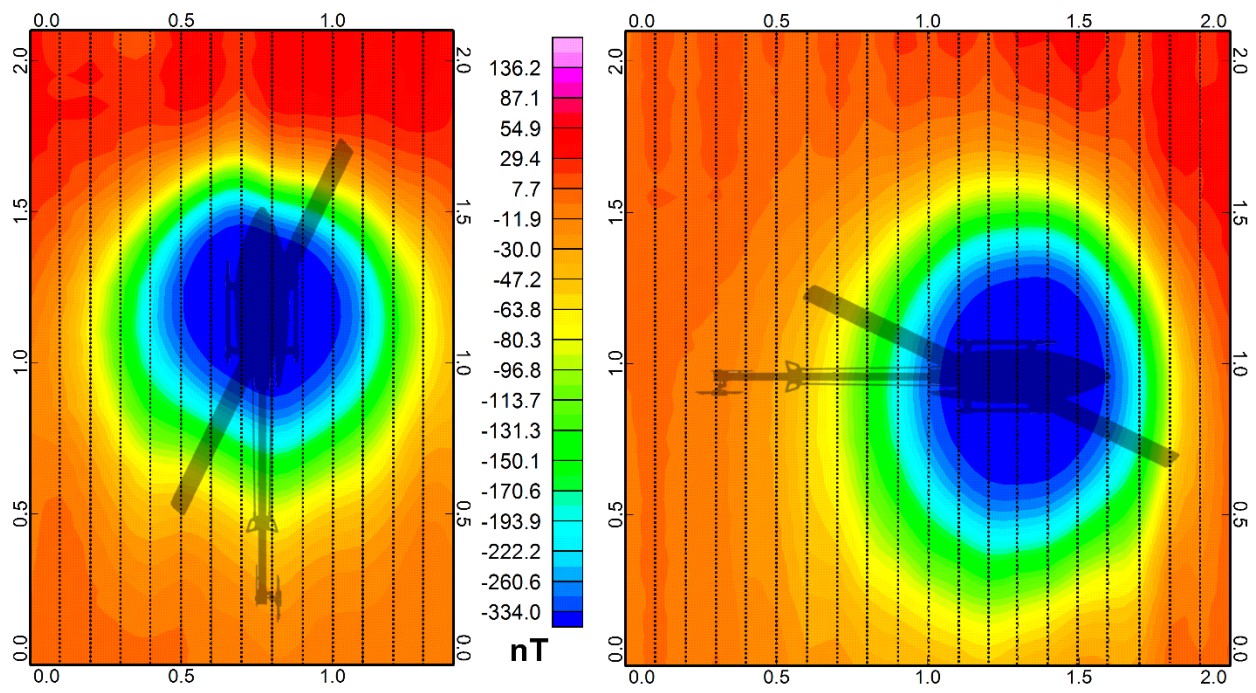

**Figure 5: Interference map of the SRH facing magnetic north (left) and east (right). Border units are in metres.**


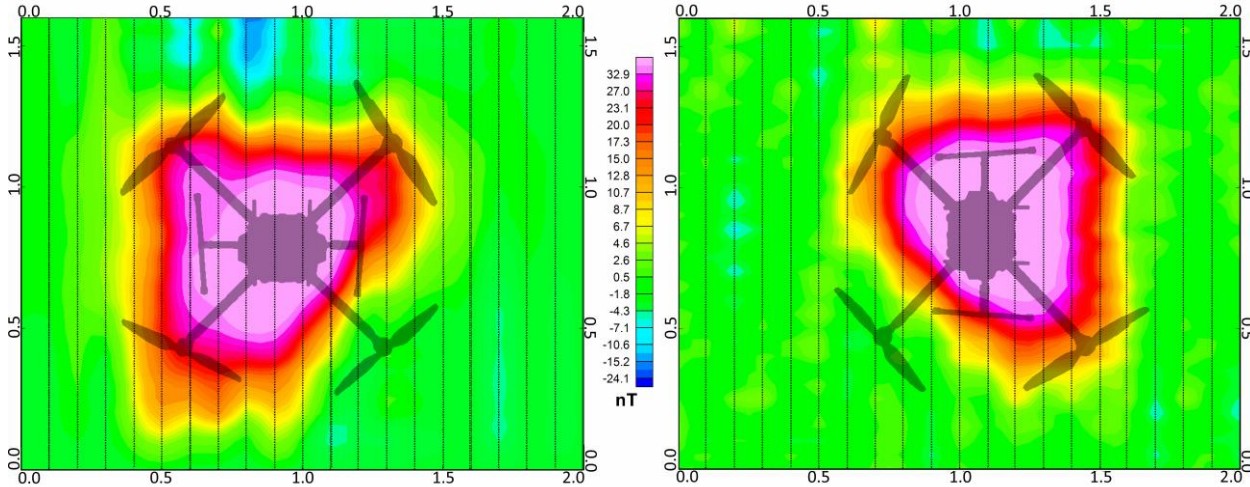

**Figure 6: Interference map of the QRH facing magnetic north (left) and east (right). Border units are in metres.**

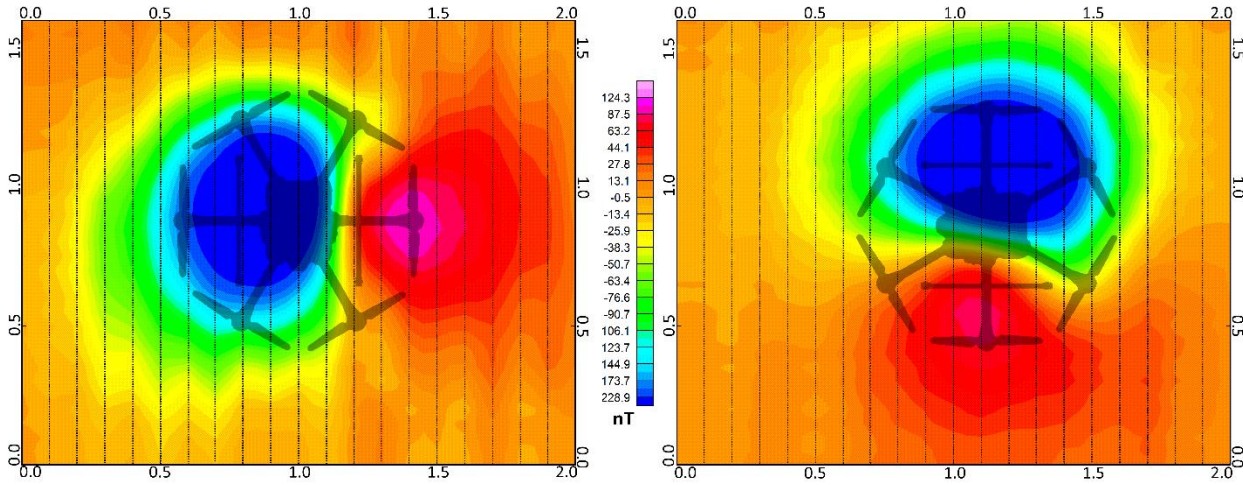

**Figure 7: Interference map of the HRH facing magnetic north (left) and east (right). Border units are in metres.**


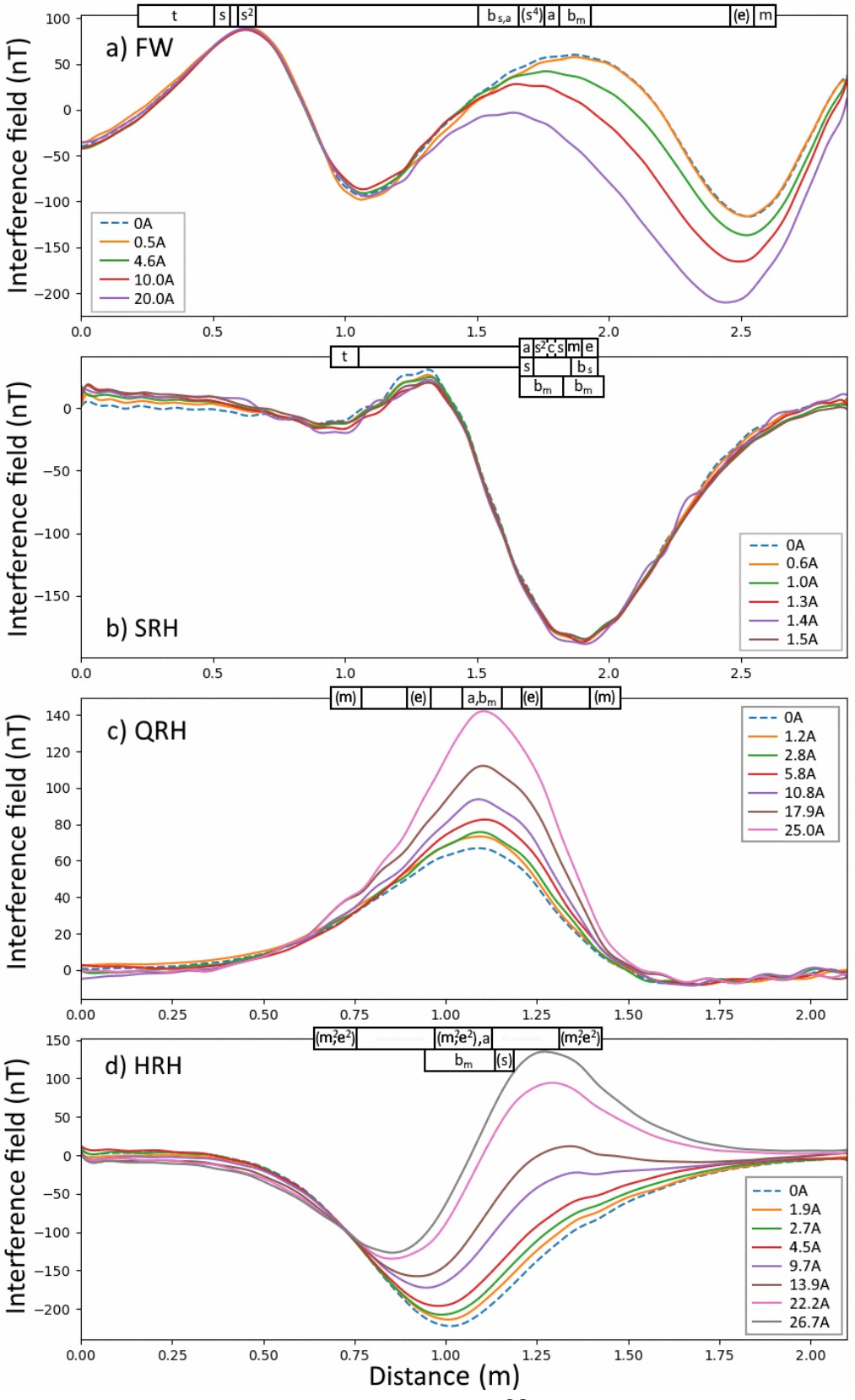

**Figure 8: The interference profiles for each UAS oriented in the north direction at different motor current draws. Profiles have been filtered using a 0.25 Hz low-pass filter. A profile was calculated for 0 A (dashed profile). The location of the UAS system components are marked on the top border of each profile set. Components are denoted as a: avionics controller/receiver, m: motor, s: servo, t: tail, c: SRH centre mast, $b_m$: motor battery, $b_s$: servo battery, $b_a$: the autopilot battery, e: ESC. Brackets indicate components are located laterally with respect to the profile axis.**

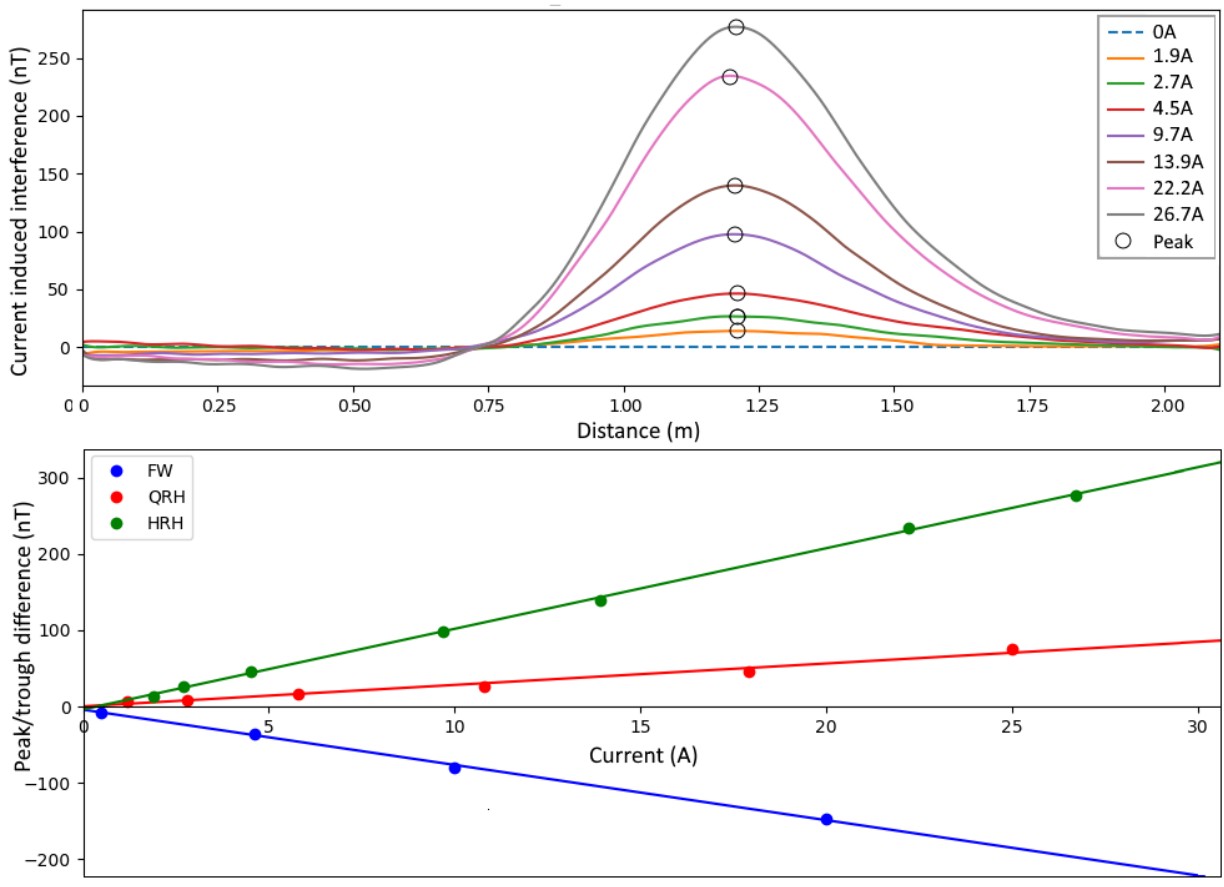

**Figure 9: (Top) The interference profiles for the HRH (with the residual removed) minus the calculated interference profile for a current of 0 A. Circles indicate current-related peak values are plotted with respect to current in bottom plot. (Bottom) A plot of the peak/trough values minus the calculated 0 A peak/trough for the FW, QRH and HRH.**
