# Peer review of "Magnetic interference mapping of four types of unmanned aircraft systems intended for aeromagnetic surveying"

_Geoscientific Instrumentation, Methods and Data Systems, 2020_

## Referee Comment (RC1) · Anonymous Referee #1 · 11 Jan 2021

Reviewer comments in yellow:

General comments: I think it's critical that the author clearly states what's actually new in this study – except for just mapping four drones. The scanner has already been used in a recent study. Also, other studies have already carried out this semi-automatic mapping of drones. So I don't really see any critical new information in this study. To be honest, this study seems to me like a fast publication in mind - but without spending proper time on providing a new method or interesting new results. Please re-think this study accordingly. Also, the Discussion needs to be far more substantial that what is presented. If the method and approach is new, the authors should be able

to properly discuss pros and cons of the system against other studies etc. as well as improvements.

Line 131: How do you filter way 60Hz background noise when mapping with much lower frequency using a GSMP35U magnetometer?

Line 155: Does this imply that the minimum noise level of any data collected over a drone is 4.2-1.1 nT? Why not carry out these measurements in the open and remove this source of noise?

160: Is it not possible to fixate the TF mag to a rod instead to remove any swaying error?

265: You map 30cm above the UAS but typically a magnetometer is just below the center of gravity. Would you then downward continue the anomaly map, to what level (if staying above the sources according to field theory) and what about the noise when downward continuing. I can't really see how a single map 30cm above the UAS provides the full answer – at least not if the ultimate goal is to achieve industry standard noise levels well below 1nT, which is needed if drones are to be used extensively.

Comments related to the method presented: o Line 160: Pendulum swing of the TF magnetometer is mention and justified by air turbulence. At such reduced travelling speeds, the aerodynamics should be negligible. The swing movement could be better described, is it perpendicular or parallel to the rail/track? Other reasons for the swing could be lack of rigidity of the set-up, the accelerations on the beginning of each line or even the amplification of motor vibrations. This could be a possible improvement on the set-up. o Line 165: Is there information loss with the application of a 0.25Hz cutting frequency filter to the data? Was the high frequency signal also present in the background lines? As the goal is to map as accurately as possible each UAV one could consider attempting to further improve the grid and reduce the need of filtering the data. o General: the GSMP-35 has the capability of sampling at 20Hz. Why was such capability not employed? o Line 209: The study/map of the SRH does not add

valuable information to the publication. So I would consider not including it. A possible way to load the engine could be the addition of a low radius drag blades just to load the motor, resulting in a more meaningful current and mapping. o Line 118: Why were two step motor used when there is only movement along one rail? Could this be simplified with the use of a single motor? Other comments o Line 217 and 226: How do the 10A and 5A current used for the standard test relate with the real operation of the UAV. It could be nice to mention how these relate to flying conditions, e.g. is 10A hovering with a specific payload? is 5A half throttle in leveled flight? o General UAV: I understand that the main focus of the publication is the method description but it could be nice to have some further details on the UAVs used. Such as weight, payloads, location of the diverse components on each UAV. This would add on the previous point about the current load. o Line 206: Could there be a confusion regarding with "Flaperons and Ailerons"? In common UAV with two actuators on each wing they are used as "Flaps and Ailerons". I would be surprised if the actuators near the root of the wing are used as Flaperons in addition to the ailerons for roll actuation. o General: Specifically for the FW UAV it could be interesting to explore additional variables in the future, such as aileron, flap, elevator and rudder deflections. For a FW UAV these are permanently being adjusted during flight. In a wing tip magnetometer setup the ailerons are the closest actuator, it is therefore interesting to understand how such deflections change the UAV signature.

---

## Referee Comment (RC2) · Anonymous Referee #2 · 28 Jan 2021

General comments

A very well drafted manuscript. It provides a good and detailed guide for those who would like to develop a lightweight UAV-borne aeromagnetometry system.

Specific comments

1. The caption of Figure 2, what are the magnetometer computer and the fluxgate computer? I know What they are for. But a bit confusing.

---

## Short Comment (SC1) · 2 Feb 2021

Dear Anonymous Referee 1,

Thank you for your review and for your comments and review.

Each of your comments have added to the quality of this paper. We have answered your comments in **bold** below.

Thank you,

Loughlin Tuck

[Figure]

Comments:

1. I think it's critical that the author clearly states what's actually new in this study – except for just mapping four drones. The scanner has already been used in a recent study. Also, other studies have already carried out this semiautomatic mapping of drones. So I don't really see any critical new information in this study. Also, the Discussion needs to be far more substantial that what is presented. If the method and approach is new, the authors should be able to properly discuss pros and cons of the system against other studies etc. as well as improvements.

**The scanner was previously used to map a single ground vehicle and the method and apparatus was accepted for publication (Hay et al., 2018). In contrast to this investigation, this manuscript maps the interference of four different aerial systems, in two directions, and measures electronic current profiles at different current draws. In addition and in contrast to other publications, it is semi-automated, it is low-cost and employs the geophysical survey system to be installed, it is performed indoors, performed safely, and is a robust method that can be used on many different forms of UAS (lines 85-91). As well, the changing of orientation of the vehicle provides information on how induced magnetism affects the overall interference signature. The varying of electronic current (not just on or off) also provides a quantified measure of its overall interference. Some of these tests are new to UAS and others have been accepted for publication, but what is new is the combination of this information as a total method. It provides a more complete picture of the magnetic interference produced and the demonstration of the method on these four vehicles provide a unique comparison that has not been addressed in literature.**

**In order to improve clarity, we changed line 96 to read: "... a demonstration of the scanner on UAS..." and the discussion has been added to on in multiple places including lines relevant to the above explanation (lines 284-289): "This**

**paper presents a quick and pragmatic method for mapping the low-frequency magnetic interference sources of a UAS in a laboratory setting. In contrast to other interference investigations, this paper presents a method that allows the UAS to be powered and running while data is collected in a semi-automated fashion that increases the maps accuracy and safety. The mappings are in two directions in order to discriminate induced interference, and profiles measures interference at different current draws. This provides a more complete picture of the low-frequency magnetic interference produced by UAS sources."**

Specific comments:
Line 131: How do you filter way 60Hz background noise when mapping with much lower frequency using a GSMP35U magnetometer?
**The manufacturer of the GSMP35U did not wish to comment on their internal processing but from our previous work, we have shown that interference from the propeller at frequencies higher than the Nyquist frequency have little effect on the measurements for the GSMP35U (Tuck et al., 2018). Therefore we suspect that there is internal filtering applied that would reduce the interference from 60 Hz sources at our 10 Hz sample rate.**
**Instead, we treat the measurements "as-is" at 10 Hz from the magnetometer. We have amended lines 131-138 to read "Interference at frequencies above the Nyquist frequency (5 Hz), such as 60 Hz electrical interference, are assumed to be aliased into the measurements. Previous work with the GSMP-35U magnetometer suggests that internal processing may apply filtering that could reduce interference aliasing (Tuck et al., 2018). Other magnetometers may have the ability to sample at higher rates that can accommodate anti-aliasing filters and reduce this interference."**

Line 155: Does this imply that the minimum noise level of any data collected over a drone is 4.2-1.1 nT? Why not carry out these measurements in the open and

remove this source of noise?

**Effectively, yes, but it is an estimate (line 158). These two values are the average and standard deviation of the root-mean-squared difference (RMSD) between all codirectional background total field (TF) measurements and those of the first codirection background line for all 8 mappings (line 158-160). Alternatively, the AVG and STD of the RMSD value in Table 2 would provide a minimum noise level for each mapping. We suggest it only as an estimate as the background subtraction was the controlling noise contributor. To clarify, we changed line 276-278 to read: "The largest contribution to the mapping error was the background subtraction and the result of lines with a high "parking" error within the magnetic gradient of the laboratory.**

**This experiment could have been conducted outdoors, but there are many reasons why readers would want to perform these experiments indoors (logistically less complex, expensive equipment at risk, etc.). This study is useful to those who wish to do magnetic noise investigations inside.**

160: Is it not possible to fixate the TF mag to a rod instead to remove any swaying error? The TF magnetometer was fixed to a rigid boom (line 113).
**The swaying was due to the carriage rocking perpendicular to the track. Line 166 is changed to read: ". . . (2) pendulum swing perpendicular to the track of the TF magnetometer due to air turbulence. . .".**

265: You map 30cm above the UAS but typically a magnetometer is just below the center of gravity. Would you then downward continue the anomaly map, to what level (if staying above the sources according to field theory) and what about the noise when downward continuing. I can't really see how a single map 30cm above the UAS provides the full answer – at least not if the ultimate goal is to achieve industry standard noise levels well below 1nT, which is needed if drones are to be used extensively.
**The interpretation is similar to that of regular aeromagnetic surveying (line 106)**

where an aircraft is flown over magnetic geology. In the case of this experiment, potential sources and their locations are known. With this information there are a few options: a) remove or mitigate the source, and/or b) model the sources using field theory (line 282).

o Line 160: Pendulum swing of the TF magnetometer is mention and justified by air turbulence. At such reduced travelling speeds, the aerodynamics should be negligible. The swing movement could be better described, is it perpendicular or parallel to the rail/track? Other reasons for the swing could be lack of rigidity of the set-up, the accelerations on the beginning of each line or even the amplification of motor vibrations. This could be a possible improvement on the set-up.
**The propellers were engaged creating significant air turbulence. Swing was perpendicular and Line 166 is changed to reflect the direction of swing.**

o Line 165: Is there information loss with the application of a 0.25Hz cutting frequency filter to the data? Was the high frequency signal also present in the background lines? As the goal is to map as accurately as possible each UAV one could consider attempting to further improve the grid and reduce the need of filtering the data.
**Only the current profiles were filtered with a 0.25Hz cut-off low-pass filter (line 174). Power spectra of the background and of the UAS measurements do not show significant power above 0.25Hz and the time-series did not show any correlation.**

o General: the GSMP-35 has the capability of sampling at 20Hz. Why was such capability not employed?
**We chose 10 Hz sampling as this provided a reasonable amount of samples for the smallest anomaly we would expect to see generated by the UAS. Using a general approach for estimating the anomaly size (Smellie, 1967), we would**

**expect a minimum anomaly full-width half-maximum of 21.5 cm or a minimum of about 20 samples. We calculated 10 Hz to already be over sampling and 20 Hz would not add any information.**

o Line 209: The study/map of the SRH does not add valuable information to the publication. So I would consider not including it. A possible way to load the engine could be the addition of a low radius drag blades just to load the motor, resulting in a more meaningful current and mapping.

**The mapping of the SRH provides information on the location as well as the intensity of that interference. We agree that the SRH current profile does not show the trend we had hoped to measure. Efforts were made to load the SRH for the current profiles but could not be accomplished safely before the return of the UAS to its owner. The current profile does show that the profiles are consistent and repeatable under small changes of current and is used to link the anomalies with sources on the airframe and shows that there is minimal electrical interference at low current. We take your recommendation to remove the SRH study under advisement.**

o Line 118: Why were two step motor used when there is only movement along one rail? Could this be simplified with the use of a single motor?

**The second step motor was added to maintain consistent travel of the carriage and avoid belt slippage. A sentence was added on line 118 to explain this: "The second stepper motor was added to avoid belt slippage and for additional torque to assure consistent speed of the carriage."**

o Line 217 and 226: How do the 10A and 5A current used for the standard test relate with the real operation of the UAV. It could be nice to mention how these relate to flying conditions, e.g. is 10A hovering with a specific payload? is 5A half throttle in leveled flight?

**The UAS motors are powered with low current during the mappings. This current is below that of the average current of each UAS during flight. We chose to power with low current for two reasons: 1. So that the permanent magnets in the outrunner motor are not stationary and create an additional variable that controls interference that does not apply during flight where motors are continuously spinning. 2. At high current, the UAS is dominated by current interference and the additional ferromagnetic interference sources would not be noticeable.**
**Our results follow Ampere's Law very well and this would suggest this current-interference relationship could be extended to higher currents experienced during flight. The following was added to the discussion (lines 321-324): "Currents of 40 A or more can be expected during flight for each of these UAS making current interference, without any mitigation, the dominant source of interference during flight."**

o General UAV: I understand that the main focus of the publication is the method description but it could be nice to have some further details on the UAVs used. Such as weight, payloads, location of the diverse components on each UAV. This would add on the previous point about the current load.
**The position of the components on each UAV are illustrated in Figure 8. The payload is included in Table 1. The table is populated with the required information for this paper but we added references to the manufacturer's specifications in Table 1 should the reader require further information.**

o Line 206: Could there be a confusion regarding with "Flaperons and Ailerons"? In common UAV with two actuators on each wing they are used as "Flaps and Ailerons". I would be surprised if the actuators near the root of the wing are used as Flaperons in addition to the ailerons for roll actuation.
**Our mistake. All references to "Flaperons" have been changed to "Flaps".**

o General: Specifically for the FW UAV it could be interesting to explore additional variables in the future, such as aileron, flap, elevator and rudder deflections. For a FW UAV these are permanently being adjusted during flight. In a wing tip magnetometer setup the ailerons are the closest actuator, it is therefore interesting to understand how such deflections change the UAV signature.

**We agree. One investigation of this has been addressed in our previous work on a fixed wing UAS (Tuck et al., 2018) but also in (Sterligov  Cherkasov, 2016).**

**References**

**Hay, A., C. Samson, L. Tuck, and A. Ellery, Magnetic surveying with an unmanned ground vehicle, Journal of Unmanned Vehicle Systems, 6 (4), 249–266, 2018.**

**Smellie, D. W., Elementary approximations in aeromagnetic interpretation, Mining Geophysics, 2 , 474–489, 1967.**

**Sterligov, B., and S. Cherkasov, Reducing magnetic noise of an unmanned aerial vehicle for high-quality magnetic surveys, International Journal of Geophysics, 2016 , 1–7, DOI: 10.1155/2016/4098275, 2016.**

**Tuck, L., C. Samson, J. Laliberté, M. Wells, and F. Bélanger, Magnetic interference testing method for an electric fixed-wing unmanned aircraft system (UAS), Journal of Unmanned Vehicle Systems, 6 (3), 177–194, DOI: 10.1186/s13023-015-0362-2, 2018.**

---

## Short Comment (SC2) · 2 Feb 2021

Dear Anonymous Referee #2,

Apologies if you are getting this twice, I do not see my original reply.

(Again), thank you for your review and for your comment.

(Specific comment 1.) As the term "computer" was vague in Fig. 2, we replaced the term "computer" and "microcomputer" with "data acquisition system" (DAS) on line 110 and 434/435. The total field and fluxgate vector magnetometer were recorded to separate DAS as the GSMP-35 DAS did not have the ability to record analog vector

magnetometer measurements.

We will provide an updated manuscript soon that reflects these changes along with corrections that address the comments of Referee #1.

Thank you,

Loughlin Tuck

---

## Author Comment (AC1) · 16 Feb 2021

Dear Anonymous Referee #1 and #2,

Thank you for your reviews and for your comments.

Each of your comments has added to the quality of this paper. It appears we had a
second account from which we have addressed each of your comments as a "short
comment" reply.

We apologize for the confusion and thank you again for your feedback,

Loughlin Tuck